# SSLA: A Generalized Attribution Method for Interpreting Self-Supervised Learning without Downstream Task Dependency

## Abstract

Self-Supervised Learning (SSL) is a crucial component of unsupervised tasks, enabling the learning of general feature representations without the need for labeled categories. However, our understanding of SSL tasks remains limited, and it is still unclear how SSL models extract key features from raw data. Existing interpretability methods are heavily reliant on downstream tasks, requiring information from these tasks to explain SSL models. This reliance blurs the line between interpreting the SSL model itself and the downstream task model. Moreover, these methods often require additional samples beyond the target of interpretation, introducing extra information that complicates the interpretability process. In this paper, we propose three fundamental prerequisites for the interpretability of SSL tasks and design the Self-Supervised Learning Attribution (SSLA) algorithm that adheres to these prerequisites. SSLA redefines the interpretability objective by introducing a feature similarity measure, reducing the impact of randomness inherent in SSL algorithms, and achieving more stable interpretability results. Additionally, SSLA abstracts the interpretability process, making it independent of specific neural network architectures. To the best of our knowledge, SSLA is the first SSL interpretability method that does not rely on downstream tasks. We also redesign a more reasonable evaluation framework and establish baselines for comparative assessment. The source code for our implementation is publicly available at `https://anonymous.4open.science/r/SSLA-EF85`.

## 1 Introduction

The advent of the big data era has brought about an abundance of unlabeled data, presenting both challenges and opportunities for the field of machine learning (Gandomi & Haider, 2015; L'heureux et al., 2017). To harness the potential of these vast unlabeled data resources, researchers have proposed the Self-Supervised Learning (SSL) paradigm (Jing & Tian, 2020). SSL leverages cleverly designed pretext tasks to enable models to automatically extract meaningful feature representations from unlabeled data, thereby improving performance on downstream tasks. This learning approach has demonstrated immense potential in various domains, including computer vision, natural language processing, and speech recognition (Liu et al., 2021; Kolesnikov et al., 2019).

In computer vision, SSL has been applied to tasks such as image classification, object detection, and semantic segmentation (Doersch et al., 2015; Kolesnikov et al., 2019). In natural language processing, it has provided new avenues for language model pretraining, text classification, and machine translation (Conneau & Lample, 2019). In the realm of speech recognition, SSL contributes to enhancing the robustness and generalization capabilities of acoustic models (Schneider et al., 2019). Furthermore, SSL is progressively showcasing its unique advantages in emerging research areas like multi-modal learning, graph neural networks, and time series analysis. Notably, SSL algorithms have also given rise to multi-modal feature extraction methods such as CLIP, which has become an essential tool for modality unification (Radford et al., 2021).

Despite the significant advantages of SSL in leveraging unlabeled data, reducing labeling costs, and enhancing model generalization, the interpretability of SSL tasks remains relatively unexplored (Guo et al., 2020; Cabannes et al., 2023). Interpretability research aims to unveil the reasons

and evidence behind model decisions, thereby increasing model transparency and trustworthiness. Attribution methods are particularly crucial in the context of SSL tasks. Compared to traditional interpretability methods, attribution methods can establish a correspondence between model outputs and individual dimensions of input features, allowing for a more granular explanation of the decision-making process (Sundararajan et al., 2017; Pan et al., 2021; Zhu et al., 2024c;b;a).

The absence of human annotations in SSL tasks renders the model's learning process and decision basis more obscure, potentially harboring biases and risks. By delving into the internal mechanisms of SSL models through attribution and other interpretability methods, we can gain a deeper understanding of how models learn from unlabeled data, identify potential biases and vulnerabilities, and ultimately improve model controllability and safety (Han et al., 2021). Moreover, interpretability research can help uncover potential flaws in SSL task design, guiding the development of more effective and robust SSL algorithms, thereby further advancing the development and application of SSL.

Current SSL interpretation methods are often confined to specific downstream tasks or rely on particular model architectures (Gur et al., 2021). When interpretations are based on specific downstream tasks, information from these tasks is inevitably introduced, leading to conceptual confusion in distinguishing whether we are interpreting the downstream task or the SSL task itself. Furthermore, interpretations tailored to specific downstream tasks tend to be limited in their applicability to other tasks. This contradicts the original intention of SSL methods, which is to extract general-purpose features, and fails to explain why SSL tasks can produce such excellent feature representations. Similarly, interpretation methods that depend on specific structures (e.g., Transformers) cannot account for the success of encoders based on other architectures, such as convolutional neural networks. Given that SSL tasks are designed with an abstraction of the encoder structure, an ideal interpretation algorithm should also possess sufficient abstraction to be broadly applicable across various model architectures.

We observe that the essence of SSL tasks lies in finding invariant features amidst variations. For instance, an encoder should extract similar features when a sample undergoes various transformations that do not alter its semantic information. An encoder with such capability can, to some extent, capture the semantic meaning of the sample. Based on this observation, we design an evaluation function $S(x, f_{\theta, Z})$ (specific details and symbol definitions are provided in Section 3.3), which is used to assess whether a sample has been well-trained. This function balances the influence of randomness during SSL task training and reflects the model's ability to identify invariance amidst variations. Subsequently, we propose the complete SSLA algorithm. SSLA can comprehensively capture the changes in $S(x, f_{\theta, Z})$ caused by variations in the sample $x$, thereby observing the contribution of each feature of the sample to the SSL model's ability to extract semantic information, and ultimately obtaining attribution results. Our contributions are summarized as follows:

1. We summarize three prerequisites for SSL task interpretation, reconstruct the objective of SSL tasks, and design the attribution method SSLA. To the best of our knowledge, SSLA is the first SSL interpretation algorithm that does not rely on any downstream tasks.

2. We systematically analyze the existing problems in the evaluation of SSL interpretation algorithms and propose a more reasonable evaluation method for interpretability.

3. We provide open-source code to facilitate reproducibility and further research by the community.

## 2 RELATED WORKS

### 2.1 SELF-SUPERVISED LEARNING

SSL aims to learn representations from large amounts of unlabeled data without the need for manual annotations, which can then be utilized to facilitate the training of downstream tasks. As a popular approach in SSL, contrastive learning (CL) learns representations by minimizing the distance between different augmented views of the same image, while maximizing the distance between views of different images. SimCLR proposes a simple contrastive learning framework that achieves comparable performance to supervised learning without using specialized negative pairs (Chen et al., 2020). MoCo-v3 employs a momentum encoder and a queue to maintain a large number of negative

samples, further improving the performance of contrastive learning (Chen et al., 2021). BYOL introduces a self-distillation-based contrastive learning method that can learn effective representations without negative pairs (Grill et al., 2020). SimSiam further simplifies the BYOL framework and demonstrates through experiments the importance of stop-gradient operations in preventing model collapse (Chen & He, 2021).

In addition to contrastive learning, masked image modeling (MIM) has emerged as another significant branch of SSL, exemplified by MAE (He et al., 2022), which achieves scalable visual representation learning by randomly masking image patches and reconstructing the missing parts. MIM methods heavily rely on the Transformer architecture, and their strong performance can be partly attributed to the self-attention mechanism of Transformers. However, our method focuses on the interpretability of general self-supervised methods that do not depend on specific model architectures or parameters. Nevertheless, we also present experimental results on MAE, which demonstrate remarkable effectiveness, showcasing its capability in visual representation learning.

## 2.2 INTERPRETABILITY OF TRADITIONAL MODELS

Several interpretability methods based on Shapley value, such as SHAP, exist for traditional models (Zhou et al., 2021). While theoretically appealing, they often fall short in practical applications (Lundberg & Lee, 2017). Particularly when dealing with complex models like deep neural networks, their computational overhead becomes prohibitive, hindering their real-world use. RISE, on the other hand, assesses feature importance by randomly masking input features and observing the changes in model output (Petsiuk et al., 2018). However, this method also faces computational bottlenecks when handling high-dimensional inputs such as images. Moreover, these methods encounter difficulties when interpreting SSL tasks. The inherent randomness in SSL training and the absence of explicit downstream task labels make it challenging for these methods to provide accurate attributions for the model's representational capabilities.

Attribution techniques play a crucial role in the interpretability research of deep learning models, aiming to reveal the contribution of input features to the model's prediction results. Compared to other interpretability methods, attribution methods adhere to two important axioms: Sensitivity and Implementation Invariance (Sundararajan et al., 2017). The Sensitivity axiom emphasizes that if an input differs from the baseline in only one feature, leading to a different prediction outcome, then this differing feature should be assigned a non-zero attribution. The Implementation Invariance axiom requires that two functionally equivalent neural networks, despite potentially different implementations, should have consistent attribution results. These axioms provide essential guidance for the design of attribution methods, helping to ensure the rationality and reliability of the attribution results.

However, most current attribution methods (Pan et al., 2021; Hesse et al., 2021; Erion et al., 2021; Kapishnikov et al., 2021; Zhu et al., 2024c;b;a) face applicability challenges in SSL tasks. Firstly, these methods are typically task-specific, requiring design tailored to particular downstream tasks (e.g., classification). The characteristic of SSL is its pre-training stage's independence from specific task labels, making it difficult to directly apply these methods. Secondly, the SSL training process often involves a large number of random operations, such as data augmentation and negative sample selection in contrastive learning. Existing attribution methods have limited capabilities in handling such randomness, making it difficult to provide stable attribution results. Therefore, given the unique nature of SSL tasks, developing attribution methods that can analyze the model's representation learning ability without specific task labels and handle the randomness in the training process is a research topic of significant importance.

## 2.3 INTERPRETABILITY OF SSL

Despite the fact that SSL does not rely on specific task labels during the pre-training phase, research on the interpretability of SSL models is still in its infancy. The Attribution Guided Factorization (AGF) method achieves visual interpretation of both supervised and SSL models by combining gradient and attribution information (Gur et al., 2021). However, when interpreting SSL tasks, this method requires the incorporation of specific downstream tasks. This dependency confines the attribution results to the specific task, failing to reflect the general-purpose features extracted by SSL models, and thus creating a conflict with the objectives of SSL. On the other hand, a hierarchical

latent variable model has been employed for in-depth interpretability analysis of MAE (Kong et al., 2023). Nevertheless, this method is primarily applicable to Transformer architectures and cannot explain the excellent performance achieved by current non-Transformer SSL models. Therefore, developing interpretation methods that are applicable to various SSL model architectures and can effectively explain the representation learning ability of SSL models without relying on specific downstream tasks remains a crucial challenge that needs to be addressed.

# 3 METHODS

## 3.1 PROBLEM DEFINITION

Let $f_\theta : \mathbb{R}^n \to \mathbb{R}^d$ denote the encoder learned through SSL, which transforms an input sample $x \in \mathbb{R}^n$ into a latent feature representation $z \in \mathbb{R}^d$, where $d \ll n$. This process is expressed as $z = f_\theta(x)$, with $\theta$ representing the parameters of the encoder.

In practical applications, the extracted latent feature $z$ is fed into a downstream task $g_\phi : \mathbb{R}^d \to \mathbb{R}^m$. This is denoted as $\tilde{y} = g_\phi(z)$, where $\phi$ represents the parameters of the downstream task model, typically obtained by fine-tuning after $\theta$ is trained. In the case of a classification task, $m$ equal to the number of classes $c$. We utilize $g_\phi(z)$ in the evaluation of SSLA.

Furthermore, we use $\mathcal{T}$ to represent the data augmentation techniques employed during SSL training. In this paper, these techniques include color jitter, Gaussian blur, grayscale conversion, random resized crop, and solarization. It is important to note that the $\mathcal{T}$ function is stochastic in nature. $S : \mathbb{R}^d \times \mathbb{R}^d \to \mathbb{R}$ denotes the function for calculating the distance between latent feature representations.

The objective of attribution is to construct a function $A : x \times f \to \mathbb{R}^n$ that transforms the input sample $x$ into an attribution result of the same dimensionality. This process is represented as $a = A(x, f)$. Here, $a_i$ signifies the importance of $x_i$ (the $i$-th dimension) to $f_\theta$. A larger $a_i$ implies that $x_i$ is more important for $f_\theta$.

## 3.2 PREREQUISITES FOR SELF-SUPERVISED LEARNING ATTRIBUTION

Before introducing the SSLA algorithm, we outline the following three prerequisites that need to be satisfied in our attribution design process, along with the rationale behind them.

**Prerequisite 1: The interpretation of SSL should not introduce information from downstream tasks. (Avoid interference from downstream training information.)** The evaluation of SSL often involves downstream tasks. For example, a common practice in visual SSL is to freeze the parameters of $f_\theta$, introduce $g_\phi$ with only one linear layer, and train the parameters $\phi$ to observe whether $f_\theta$ can extract linearly separable features (e.g., SimCLR (Chen et al., 2020)). However, during the interpretation of SSL tasks, $g_\phi$ or its parameters $\phi$ should not be introduced. The reason is that $\phi$ is trained separately after the SSL task, and we cannot be certain whether the importance in the final attribution result $a$ comes from $g_\phi$ or $f_\theta$. Some SSL interpretability methods introduce information from $\phi$ during the interpretation process (Gur et al., 2021). More importantly, once information from $g_\phi$ is introduced, the interpretability results are limited to the downstream task, rather than providing an explanation of the SSL task itself.

**Prerequisite 2: The interpretation process should not introduce samples other than the current sample.** Similar to the reasoning behind **Prerequisite 1**, sampling additional samples introduces extraneous information. The attribution results are influenced by the sampling process and the information from the introduced samples. Moreover, the attribution results are biased towards the interaction between samples, rather than the model's focus on the current sample.

**Prerequisite 3: The interpretation process should not be restricted to specific model architectures.** SSL algorithms abstract the encoder structure during their design, which implies that SSL algorithms themselves do not heavily rely on specific model architectures. Specifically, some interpretability methods that utilize the self-attention mechanism for importance analysis cannot ex-

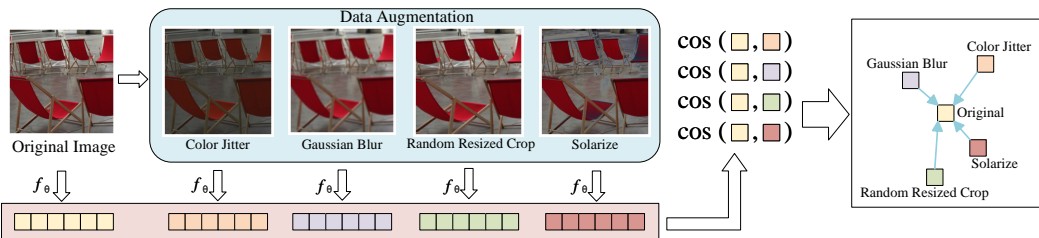

Figure 1: Similarity calculation diagram. The cosine similarity in the figure should be as close as possible.

plain why CNN-based structures also achieve excellent SSL results (Caron et al., 2021). Our SSLA algorithm also abstracts the encoder structure and cannot depend on any specific architecture.

### 3.3 SELF-SUPERVISED LEARNING ATTRIBUTION (SSLA)

The core idea of SSL is to find invariant features amidst variations. For example, after applying multiple transformations $\tau_1, \tau_2 \sim \mathcal{T}$ to an input sample $x$, we expect the encoder $f_\theta$ to extract similar feature representations. Typically, we use cosine similarity as the evaluation metric for similarity, i.e., we desire a higher value for $\cos(f_\theta(\tau_1(x)), f_\theta(\tau_2(x)))$.

Different SSL methods may have different training strategies, such as sampling negative samples during self-supervised training (e.g., SimCLR (Chen et al., 2020)) or using multiple encoders with similar functionality (e.g., BYOL (Grill et al., 2020), where one $\theta$ adopts a similar $\theta'$). This core idea can also be extended to reconstruction-based methods like MAE (He et al., 2022), where the variation lies in the random masking process, and the expectation is that the model can reconstruct the same image regardless of which regions are masked. The effectiveness of the model on a sample can be evaluated by $E_{\tau_1, \tau_2 \sim \mathcal{T}}[\cos(f_\theta(\tau_1(x)), f_\theta(\tau_2(x)))]$. However, when interpreting SSL models, we need to maintain this core idea while reducing the influence of randomness and ensuring that the interpretability algorithm can directly operate on the original sample. Therefore, we adopt $E_{\tau \sim \mathcal{T}}[\cos(f_\theta(x), f_\theta(\tau(x)))]$ for evaluation (Figure 1 provides a better illustration of this process). As shown in Figure 1, features are extracted from both the original image and its augmented versions, and then the cosine similarity between the features of these augmented images and the original image is calculated. The similarity of these features obtained from different augmentations should be as close as possible to the features of the original image.

Essentially, all transformations are based on the premise of not destroying the semantic information represented by the input, meaning that the same semantic information should yield the same output. Therefore, interpreting an SSL-learned model involves explaining the contribution of each feature to preserving the semantic information of the sample. In this process, we observe the impact of modifying each feature on $E_{\tau \sim \mathcal{T}}[\cos(f_\theta(x), f_\theta(\tau(x)))]$.

Since the purpose of $f_\theta(\tau(x))$ is to compute similarity with the original sample, it remains unchanged during feature modification, eliminating the need for repeated calculations. We can use $Z = [z_1, z_2, \cdots, z_N]$ for caching, where $z_i = \tau(x), \tau \sim \mathcal{T}$. Thus, the evaluation of whether the model is effective on a sample can be transformed into:

$$S(x, f_\theta, Z) = E_{z \sim Z}[\cos(f_\theta(x), z)] = \frac{1}{N} \sum_{i=1}^{N} \cos(f_\theta, z_i)$$

It is worth noting that many attribution methods use the model's training loss function as a tool to evaluate its effectiveness on a sample (Sundararajan et al., 2017; Pan et al., 2021; Zhu et al., 2024c;b;a). Generally, a lower loss function indicates that the sample is better learned. However, the training loss function may not be suitable for interpreting SSL tasks. Considering Prerequisites 1 and 2, we cannot use downstream tasks to explain the model. We have both the loss function of the downstream task and the loss function for training SSL. The former is not generalizable and can only be applied to a single downstream task; the latter sometimes introduces negative samples, leading

to an uncontrollable interpretation process (it is feasible when no negative samples are introduced, such as in BYOL (Grill et al., 2020) and SimSiam (Chen & He, 2021)). Our redesigned $S(x, f_\theta, Z)$, derived directly from the SSL algorithm design perspective, is more versatile as it starts from a single sample, avoiding the aforementioned issues. Moreover, since we only require the output of $f_\theta$ during the design process, independent of the structure of $f_\theta$, the design satisfies Prerequisite 3.

With $S(x, f_\theta, Z)$, the design of SSLA becomes straightforward. The core of the SSLA algorithm lies in observing the contribution of changes in different dimensions of the sample to $S(x, f_\theta, Z)$, while satisfying the two axioms proposed in attribution: Sensitivity and Implementation Invariance (Sundararajan et al., 2017). We prove the satisfaction of these two axioms by the SSLA algorithm in **Appendix B and C**. For ease of understanding, we provide some intuition: if a slight change in a feature leads to a significant deviation in the corresponding $S(x, f_\theta, Z)$, it indicates that this feature is crucial. In this process, we need to ensure that the change is very subtle, i.e., not too different from the original sample. To ensure the efficiency of the sample modification's impact on $S(x, f_\theta, Z)$, we seek the feature modification method that changes $S(x, f_\theta, Z)$ most rapidly from a first-order perspective. The proof is provided in **Appendix A**.

Next, we present the most important theorem of this paper:

**Theorem 1.** *Given an SSL-trained encoder $f_\theta$, a sample $x$, and its corresponding transformation set $Z$, we can obtain $A(x) \in \mathbb{R}^n$,*

$$A(x) = \sum_{t=1}^{T} \frac{x}{T} \cdot \left| \frac{\partial S(x_{t-1}, f_\theta, Z)}{\partial x_{t-1}} \right|$$

*as the attribution result, where $T$ represents the number of sample updates, and the update process is defined as:*

$$x_t = x_{t-1} - \frac{x}{T} \cdot \text{sign}\left( \frac{\partial S(x, f_\theta, Z)}{\partial x_{t-1}} \right)$$

*with $x_0$ being the original sample.*

Here, we provide the core steps in the derivation of the SSLA algorithm; detailed steps are presented in **Appendix B**.

$$A(x) = - \int (x_t - x_{t-1}) \cdot \frac{\partial S(x, f_\theta, Z)}{\partial x_{t-1}} dt$$

$$= \sum_{t=1}^{T} \frac{x}{T} \cdot \left| \frac{\partial S(x, f_\theta, Z)}{\partial x_{t-1}} \right| \quad (1)$$

$$\sum_{i=1}^{n} A_i(x) = S(x_0, f_\theta, Z) - S(x_T, f_\theta, Z) \quad (2)$$

As shown in equations (1) and (2), the SSLA algorithm accumulates the corresponding gradient information during the update process, where $\frac{x}{T}$ represents the learning rate for sample updates. This is illustrated in (Zhu et al., 2024d): dimensions with larger values should be explored more. We can observe that the sum of attributions across all feature dimensions equals the change in $S(x, f_\theta, Z)$, which implies that the SSLA algorithm satisfies Sensitivity Axiom (Sundararajan et al., 2017). The accumulation of all gradients for a single dimension can represent the contribution of that dimension to the change in the result in a first-order approximation, which is the corresponding attribution result. The figure 2 illustrates the iterative process of the SSLA algorithm. It begins with data augmentation of the original image, followed by encoding both the original and augmented images. Cosine similarities between the original and augmented features are calculated, and these similarities are used to update the attribution result iteratively. The process is repeated $T$ times to refine the attribution $a_i$, identifying the features most important for the SSL model's understanding. The pseudocode is shown in Algorithm 1.

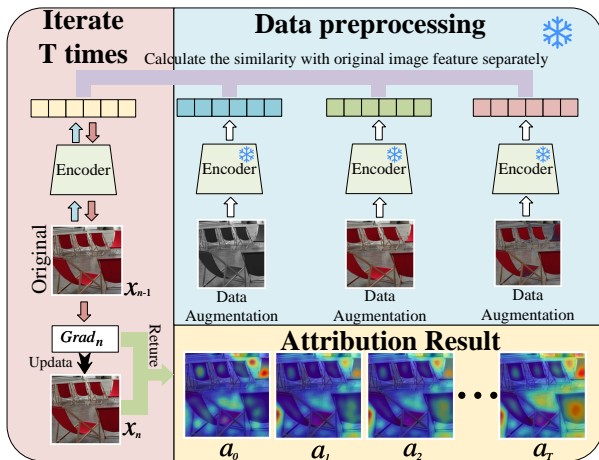

Figure 2: The Flowchart of SSLA. Light blue upward arrows indicate forward propagation, light red downward arrows indicate backward propagation, and snowflake icons represent the absence of gradient propagation.

---

**Algorithm 1** Self-Supervised Learning Attribution (SSLA)

---

**Input:** Iterate times $T$, Data Augmentation $Z$, model parameter $\theta$
**Output:** $A$
  1: **Initial:** $A = 0$, $x_0 = x$
  2: **for** $t$ in range $(1, T)$ **do**
  3:      $A = A + \frac{x}{T} \cdot \left| \frac{\partial S(x_{t-1}, f_\theta, Z)}{\partial x_{t-1}} \right|$
  4:      $x_t = x_{t-1} - \frac{x}{T} \cdot \text{sign} \left( \frac{\partial S(x_{t-1}, f_\theta, Z)}{\partial x_{t-1}} \right)$
  5: **end for**
  6: **return** $A$

---

## 4 EXPERIMENTS

### 4.1 DATASET AND MODEL

We conducted our main experiments on the ImageNet image dataset (Deng et al., 2009), using the same dataset split as MFABA (Zhu et al., 2024c). We employed the ResNet-50 (He et al., 2016) model as our experimental backbone.

### 4.2 CHOICE OF SELF-SUPERVISED LEARNING METHODS

In our experiments, we selected five representative SSL methods: BYOL (Grill et al., 2020), SimCLR (Chen et al., 2020), SimSiam (Chen & He, 2021), MoCo-v3 (Chen et al., 2021), and MAE (He et al., 2022), to test our proposed SSLA method. BYOL learns high-quality representations through bootstrapping positive sample pairs; SimCLR relies on a large number of negative samples for contrastive learning; SimSiam adopts a Siamese network architecture and trains solely on positive sample pairs, without the need for negative samples or a momentum encoder; MoCo-v3 optimizes the contrastive learning process by using a momentum encoder to generate a dynamic queue of negative samples; and MAE forces the model to learn global contextual information through masking. These methods achieve accuracies of 83.1%, 68.7%, 68.3%, 74.6%, and 85.9%, respectively, on the ImageNet-1k downstream task. These methods represent different technical paths in the SSL field. By testing our SSLA method on these diverse approaches, we can comprehensively evaluate the applicability and effectiveness of SSLA across various SSL models.

## 4.3 Evaluation Method

Traditional interpretability evaluation methods, such as Insertion score, Deletion score, and INFD score (Petsiuk et al., 2018; Yeh et al., 2019), are impractical when applied to the evaluation of interpretability methods for self-supervised tasks. The fundamental reason lies in the necessity of introducing the concept of a baseline in these traditional methods. The baseline is typically set as an all-zero image or an image with added perturbations (usually Gaussian blur) (Sturmfels et al., 2020). However, a successfully established baseline should, in essence, not provide any guidance to the current model's decision-making. That is, replacing features in the input with their corresponding values from the baseline should decrease the model's decision-making capability. However, for SSL tasks, replacing with zeros or adding perturbations to the sample has minimal impact on the decision.

As previously mentioned, the essence of SSL lies in seeking invariance through transformations. Therefore, the traditional process of replacing baselines can also be viewed as a form of transformation. For example, in a trained SSL model, operations like Gaussian blur have minimal impact on the final similarity calculation. This is because data augmentation methods such as Gaussian blur are already incorporated as part of the transformation set ($\mathcal{T}$) during SSL training. Unfortunately, there is currently no baseline that can both consistently affect SSL tasks and align with intuitive understanding. Hence, there is an urgent need to propose new evaluation methods specifically for interpretability in SSL tasks.

Our core idea is to evaluate the potential of different regions in influencing the decision-making of SSL tasks. In simple terms, if a feature is more important for the decision-making of an SSL task, then fixing this feature and modifying other features should have a smaller impact on the model's output, and vice versa. Under this assumption, we need to clarify two basic definitions for evaluation:

1. How to define the impact on the model's output.
2. How to ensure that modifying other features is definitely effective.

First, let's define the impact on the model's output. As mentioned earlier, cosine similarity is used in SSL tasks to assess the similarity of outputs, with the expectation of finding invariant features amidst variations. Following this line of thought, we can use cosine similarity to quantify the degree of change in the output. Specifically, we use $\cos(f_\theta(x), f_\theta(\tilde{x}))$ to evaluate the change in decision-making caused by transforming sample $x$ into $\tilde{x}$. A smaller value indicates a larger impact on the output. When $\tilde{x} = x$, the cosine similarity is constantly 1, indicating no impact on the output. Adopting this evaluation approach avoids the selection of a baseline, thereby eliminating the bias introduced by baseline selection in previous evaluation methods.

Next, we address the issue of ensuring the effectiveness of modifying other features. This leads to our core hypothesis:

**Theorem 2.** $\forall m \in \mathbb{R}^n$ and $m_j \in \{0, 1\}$, there exists an update direction $m \cdot \mathrm{sign}\left(\frac{\partial \cos(f_\theta(\tilde{x}), f_\theta(x))}{\partial \tilde{x}}\right)$ that guarantees the effectiveness of feature modification, i.e., $\cos(f_\theta(x), f_\theta(\tilde{x})) \leq 0$.

The mask is created by selecting the features with the highest attribution scores and setting the corresponding proportion of $m_j$ to 0. The proof of Theorem 2 is provided in **Appendix D**. To assess the model's sensitivity, we modify the features outside the masked region. If the model is less affected by these modifications, it indicates that the features within the masked region are more important; conversely, if the model is significantly affected, it suggests that the masked features are less critical.

## 4.4 Parameters and Evaluation Settings

Our method primarily involves four key parameters: $\epsilon$, which controls the magnitude of adversarial perturbations, is set to $\frac{16}{255}$; Step Number, which determines the number of iterative attack steps, is set to 10 for SimCLR, 50 for MoCo-v3 and MAE, and 70 for BYOL and SimSiam; and the learning rate, which is set to 0.01 for MAE and 0.001 for all other methods. Furthermore, to ensure experimental efficiency and fairness, we set the sample size to 1000. The Mask Rate is set to [0%, 10%, 20%, 30%, 40%, 50%, 60%, 70%, 80%, 90%, 100%], where 0% represents no masking,

Table 1: Experimental results showing the performance of SSLA across different SSL methods under varying Mask Rates. The table compares the impact of Masking Important and Masking Unimportant features on the attribution effectiveness of SSLA and Random Mask methods. Mask Important (MI) represents scenarios where key features are masked, while Mask Unimportant (MU) reflects cases where less critical features are masked. A higher value in the MI indicates better attribution by SSLA, while a lower value in the MU demonstrates SSLA's effectiveness in minimizing interference from less important features. During random masking, since we cannot determine the importance of features, we only retain one masking scenario without distinguishing between masking important or unimportant features.

| | BYOL | | | SimCLR | | | SimSiam | | | MoCo-v3 | | | MAE | | |
|---|---|---|---|---|---|---|---|---|---|---|---|---|---|---|---|
| Mask Rate | Random Mask | SSLA MI | SSLA MU | Random Mask | SSLA MI | SSLA MU | Random Mask | SSLA MI | SSLA MU | Random Mask | SSLA MI | SSLA MU | Random Mask | SSLA MI | SSLA MU |
| 0% | 0.62 | - | - | 0.56 | - | - | 0.57 | - | - | 0.53 | - | - | 0.68 | - | - |
| 10% | 0.64(±0.0099) | 0.66 | 0.64 | 0.59(±0.0113) | 0.60 | 0.58 | 0.59(±0.0077) | 0.62 | 0.59 | 0.55(±0.014) | 0.57 | 0.54 | 0.70(±0.0932) | 0.74 | 0.69 |
| 20% | 0.66(±0.01) | 0.70 | 0.66 | 0.61(±0.011) | 0.64 | 0.61 | 0.62(±0.0081) | 0.66 | 0.61 | 0.57(±0.0134) | 0.60 | 0.56 | 0.72(±0.0855) | 0.78 | 0.70 |
| 30% | 0.68(±0.009) | 0.73 | 0.68 | 0.63(±0.0109) | 0.67 | 0.63 | 0.65(±0.0073) | 0.71 | 0.63 | 0.60(±0.0122) | 0.64 | 0.59 | 0.75(±0.0796) | 0.82 | 0.72 |
| 40% | 0.71(±0.0084) | 0.77 | 0.70 | 0.67(±0.0103) | 0.72 | 0.66 | 0.68(±0.0071) | 0.76 | 0.66 | 0.64(±0.0116) | 0.69 | 0.62 | 0.78(±0.071) | 0.86 | 0.74 |
| 50% | 0.74(±0.0074) | 0.81 | 0.73 | 0.70(±0.0095) | 0.76 | 0.70 | 0.72(±0.0063) | 0.81 | 0.69 | 0.68(±0.0099) | 0.73 | 0.65 | 0.80(±0.0639) | 0.89 | 0.76 |
| 60% | 0.78(±0.0065) | 0.85 | 0.76 | 0.74(±0.0086) | 0.82 | 0.74 | 0.76(±0.0054) | 0.86 | 0.73 | 0.72(±0.0085) | 0.79 | 0.70 | 0.84(±0.0536) | 0.92 | 0.79 |
| 70% | 0.82(±0.0052) | 0.90 | 0.80 | 0.79(±0.0076) | 0.87 | 0.79 | 0.81(±0.0047) | 0.90 | 0.77 | 0.78(±0.0072) | 0.84 | 0.75 | 0.87(±0.0418) | 0.95 | 0.83 |
| 80% | 0.88(±0.004) | 0.94 | 0.85 | 0.85(±0.006) | 0.92 | 0.85 | 0.88(±0.0037) | 0.95 | 0.83 | 0.85(±0.0053) | 0.90 | 0.82 | 0.91(±0.0292) | 0.97 | 0.87 |
| 90% | 0.95(±0.0023) | 0.98 | 0.92 | 0.92(±0.004) | 0.97 | 0.92 | 0.95(±0.0024) | 0.98 | 0.91 | 0.93(±0.0031) | 0.97 | 0.90 | 0.96(±0.014) | 0.99 | 0.92 |
| 100% | 1.0(±0.0) | 1 | 1 | 1.0(±0.0) | 1 | 1 | 1.0(±0.0) | 1 | 1 | 1.0(±0.0) | 1 | 1 | 1.0(±0.0) | 1 | 1 |

100% represents masking all regions, and 10% represents masking 10% of the regions. During the evaluation, we employ the first-order adversarial attak (Goodfellow et al., 2014) to attack the unmasked regions, and then assess the similarity between the attacked samples and the original images. All experiments were conducted on a Red Hat Enterprise Linux release 8.6 (Ootpa) system equipped with an NVIDIA A100 GPU.

## 4.5 RESULTS

In this experiment, as we are the first to propose an attribution method that does not utilize downstream task information, we lack a direct baseline for comparison. We choose to compare random masking (Random Mask) with masking based on the ranking of attribution values computed by SSLA. By setting different Mask Rates, we mask both important and unimportant features and observe the changes in similarity between the attacked samples and the original images.

When important features are masked, we apply a first-order adversarial attack to the unmasked regions. A higher similarity between the attacked sample and the original image indicates a greater impact of the masked features on the model's decision-making. As shown in Table 1, the SSLA method excels in this strategy, accurately identifying features that play a crucial role in the model's decisions. Specifically, at a Mask Rate of 50%, SSLA outperforms Random Mask across all SSL methods, with the highest similarity score observed for MAE, reaching 0.89 for SSLA when masking important features, compared to 0.80 for Random Mask. Similar trends are seen for BYOL, SimCLR, SimSiam, and MoCo-v3, with SSLA achieving similarity scores of 0.81, 0.76, 0.81, and 0.73, respectively, for masking important features, while Random Mask scores are comparatively lower at 0.74, 0.70, 0.72, and 0.68.

Furthermore, when unimportant features are masked, SSLA consistently produces lower similarity scores between the attacked and original samples, demonstrating its effectiveness in reducing the influence of unimportant features. For example, in the case of MoCo-v3 at a 50% Mask Rate, the similarity score drops to 0.65 when masking unimportant features using SSLA, as opposed to 0.68 for Random Mask. Similarly, for BYOL and MAE, SSLA achieves similarity scores of 0.73 and 0.76, while Random Mask results in higher similarity scores of 0.74 and 0.80, respectively. These results validate the stability and reliability of SSLA in both highlighting crucial features and minimizing interference from unimportant regions.

Additionally, as the Mask Rate increases, the SSLA method continues to outperform Random Mask in masking important features. At a 90% Mask Rate, SSLA achieves the highest similarity score for MAE at 0.99, while Random Mask remains slightly behind at 0.96. For BYOL, SimCLR, SimSiam, and MoCo-v3, the SSLA method consistently reaches similarity scores of 0.98, 0.97, 0.98, and 0.97, respectively, compared to Random Mask scores of 0.95, 0.92, 0.95, and 0.93.

In conclusion, the SSLA method demonstrates clear superiority in both masking important and unimportant features, proving its effectiveness in accurately attributing features that influence SSL task decision-making. By offering higher similarity scores when important features are masked and lower scores when unimportant features are masked, SSLA consistently provides more reliable and stable attribution results compared to Random Mask across various SSL methods and Mask Rates.

## 5 CONCLUSION

In this work, we proposed the Self-Supervised Learning Attribution (SSLA) method, designed to address the unique challenges of interpreting SSL models. Through extensive experiments on the ImageNet dataset using a variety of SSL methods, including BYOL, SimCLR, SimSiam, MoCo-v3, and MAE, we demonstrated the robustness and applicability of the SSLA method. The evaluation results indicate that SSLA can effectively attribute features, distinguishing between those that are critical and those that are less important to the model's decision-making process. This capability is consistent across different SSL models and varying mask rates, highlighting SSLA's adaptability. Moreover, our experiments showed that SSLA outperforms random masking strategies, providing more accurate and stable attributions. The method successfully guides models to focus on the regions that genuinely influence decision-making, thereby enhancing both the interpretability and trustworthiness of SSL models.

## CODE OF ETHICS AND ETHICS STATEMENT

All authors of this paper have read and adhered to the ICLR Code of Ethics[1]. This work does not raise any ethical concerns regarding the collection, release, or use of datasets, nor does it involve human subjects. The methods proposed in this paper aim to improve the interpretability of self-supervised learning models without introducing harmful insights or biases. The algorithm designed for this study does not rely on downstream task information and ensures fairness, avoiding any conflicts of interest or sponsorship-related issues. All results presented in this paper comply with research integrity and data privacy standards.

## REPRODUCIBILITY

To ensure reproducibility, we have provided comprehensive details of our proposed SSLA algorithm in the main text, including the problem definition, methodology, and evaluation framework. The source code for the SSLA algorithm, along with instructions for data preprocessing and experiments, is made publicly available at the Anonymous Repository[2]. Further details regarding theoretical assumptions, proofs, and parameter settings are provided in the appendix. For the datasets used in this work, all preprocessing steps and experimental configurations are documented in the supplemental materials to facilitate the replication of our results.

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

## A    PROOF OF FEATURE CHANGES DIRECTION

$$S\left(x_j, f_\theta, Z\right) = S\left(x_{j-1}, f_\theta, Z\right) + \frac{\partial S\left(x_{j-1}, f_\theta, Z\right)}{\partial x_{j-1}}\left(x_j - x_{j-1}\right) + \mathcal{O}$$

The fastest descent occurs when $\left(x_j - x_{j-1}\right) = -\frac{\partial S(x_{j-1}, f_\theta, Z)}{\partial x_{j-1}}$. Additionally, when $\cos\left(\left(x_j - x_{j-1}\right), \frac{\partial S(x_{j-1}, f_\theta, Z)}{\partial x_{j-1}}\right) < 0$, it indicates that the value decreases.

## B    PROOF OF THEOREM 1 AND SENSITIVITY AXIOM

*Proof.* We start by applying a first-order Taylor expansion to $S(x_{j+1}, f_\theta, Z)$ around $x_j$, omitting the higher-order infinitesimals $\mathcal{O}$ in the process:

$$S(x_{j+1}, f_\theta, Z) \approx S(x_j, f_\theta, Z) + \frac{\partial S(x_j, f_\theta, Z)}{\partial x_j}(x_{j+1} - x_j)$$

Summing this over all $n$ steps and rearranging, we focus on the change in $S$:

$$S(x_n, f_\theta, Z) - S(x_0, f_\theta, Z) \approx \sum_{j=0}^{n-1} \frac{\partial S(x_j, f_\theta, Z)}{\partial x_j}(x_{j+1} - x_j)$$

Substituting the update rule for $x_{j+1}$ and simplifying:

$$S(x_n, f_\theta, Z) - S(x_0, f_\theta, Z) \approx \sum_{j=0}^{n-1} \frac{\partial S(x_j, f_\theta, Z)}{\partial x_j}\left(-\frac{x}{T} \cdot \text{sign}\left(\frac{\partial S(x_j, f_\theta, Z)}{\partial x_j}\right)\right)$$

$$= -\frac{1}{T}\sum_{j=0}^{n-1} x \cdot \left|\frac{\partial S(x_j, f_\theta, Z)}{\partial x_j}\right|$$

Finally, rearranging and taking the absolute value yields the attribution formula:

$$\sum_{j=0}^{n-1}\left|\frac{x}{T} \cdot \frac{\partial S(x_j, f_\theta, Z)}{\partial x_j}\right| \approx S(x_0, f_\theta, Z) - S(x_n, f_\theta, Z)$$

This demonstrates that the sum of attributions reflects the change in S, satisfying the Sensitivity axiom. Thus, we have proven both Theorem 1 and the adherence of SSLA to the Sensitivity axiom.

$\square$

## C    PROOF OF IMPLEMENTATION INVARIANCE AXIOM

The Self-Supervised Learning Attribution (SSLA) algorithm adheres to the chain rule. Based on the properties of gradients, the SSLA algorithm satisfies implementation invariance, ensuring that results are consistent across different valid implementations of the same functional relationship.

## D    PROOF OF THEOREM 2

*Proof.* From the first-order Taylor expansion, we have:

$$\cos\left(f_\theta(\tilde{x} + \Delta x), f_\theta(x)\right) \approx \cos\left(f_\theta(\tilde{x}), f_\theta(x)\right) + \Delta x^\top \cdot \frac{\partial \cos\left(f_\theta(\tilde{x}), f_\theta(x)\right)}{\partial \tilde{x}}$$

Rearranging the terms, we get:

$$\cos\left(f_\theta(\tilde{x}+\Delta x),f_\theta(x)\right)-\cos\left(f_\theta(\tilde{x}),f_\theta(x)\right)\approx\Delta x^\top\cdot\frac{\partial\cos\left(f_\theta(\tilde{x}),f_\theta(x)\right)}{\partial\tilde{x}}$$

If $\Delta x^\top\cdot\frac{\partial\cos(f_\theta(\tilde{x}),f_\theta(x))}{\partial\tilde{x}}\leqslant 0$, then we have:

$$\cos\left(f_\theta(\tilde{x}+\Delta x),f_\theta(x)\right)\leq\cos\left(f_\theta(\tilde{x}),f_\theta(x)\right)$$

Incorporating the mask $m$, where

$$m_j=\begin{cases}0 & \text{if masked}\\1 & \text{else}\end{cases}$$

we define the change $\Delta x$ as:

$$\Delta x=-\operatorname{sign}\left(\frac{\partial\cos(f_\theta(\tilde{x}),f_\theta(x))}{\partial\tilde{x}}\right)\cdot m$$

Substituting this into the previous inequality, we obtain:

$$-m\cdot\operatorname{sign}\left(\frac{\partial\cos(f_\theta(\tilde{x}),f_\theta(x))}{\partial\tilde{x}}\right)\cdot\frac{\partial\cos\left(f_\theta(\tilde{x}),f_\theta(x)\right)}{\partial\tilde{x}}=-\sum m_j\cdot\left|\frac{\partial\cos\left(f_\theta(\tilde{x}),f_\theta(x)\right)}{\partial\tilde{x}_j}\right|\leqslant 0$$

This demonstrates that even after incorporating the mask, we can still ensure the effectiveness of feature modification. □

