# OpenReview forum: "SSLA: A Generalized Attribution Method for Interpreting Self-Supervised Learning without Downstream Task Dependency"
_ICLR.cc/2025/Conference — Submitted to ICLR 2025_

### Official Review · Reviewer_oq2z · 2024-10-25

**Soundness:** 2
**Presentation:** 3
**Contribution:** 2
**Rating:** 5
**Confidence:** 4

**Summary:**

This paper indicates that the introduced additional samples from downstream task would impede the interpretability of Self-Supervised Learning (SSL). To tackle this issue, the authors try to propose a new interpretability objective by introducing a feature similarity measure, decoupling the interpretability process from the reliance of downstream tasks.

**Strengths:**

- The additional information introduced by prediction head and downstream tasks may influence the interpretability of SSL is a crucial and challenging  breakthrough point.
-  Three fundamental prerequisites for the interpretability of SSL proposed by this paper sounds reasonable.

**Weaknesses:**

- Despite the author indicating the potential weakness for the interpretability of SSL, the alignment ability for data augmentation is just one side of current self-supervised learning. The other component, which is often regarded as an extra design to prevent training collapse intuitively, is essentially to ensure that the representation divergence is sharp enough to cluster the data distribution by latent categories. More details can be found in [1] and [2]. Therefore, only adopting the extent of augmentation that is invariant to evaluate the related influence of variables is quite biased.

   - [1] Awasthi, Pranjal et al. “Do More Negative Samples Necessarily Hurt in Contrastive Learning?” International Conference on Machine Learning (2022).
   - [2] Weiran Huang, Mingyang Yi, Xuyang Zhao, and Zihao Jiang. Huang, Weiran, et al. "Towards the generalization of contrastive self-supervised learning." arXiv preprint arXiv:2111.00743 (2021).

**Questions:**

**Reviewing summary**
- As listed in the weaknesses, I think the authors did not incorporate the divergence into their consideration, which can be regarded as the most critical component of contrastive self-supervised learning, making their criterion sound unreasonable. Despite their indicating the unreasonable aspects regarding the interpretability of SSL in relation to downstream tasks, this results in my score of 3.

---

> ### Author Response · Authors · 2024-11-19
>
> ### Response for Weakness
>
> The cited works [1] and [2] focus on training techniques for SSL tasks, which differ from our research objective of interpreting SSL models. Our method is not limited to any specific SSL training strategy and can be applied to models trained using the approaches discussed in [1] and [2]. While [2] emphasizes divergence around class centers and concentration of augmented data, these aspects primarily influence the training phase. Our study focuses on interpreting SSL models post-training, independent of the specific training techniques employed.
>
> Furthermore, as noted in line 244 of our manuscript, the sampling strategy for \(\tau\) in our method mirrors the one used during SSL training, preserving properties like class-centered divergence and augmented data concentration. We will clarify in the manuscript that our approach is equally applicable to SSL models trained with methods like those described in [1] and [2].
>
> ---
>
> ### Response for Question
>
> SSL models trained using divergence-based techniques are fully compatible with our method. Our primary focus is on interpreting SSL models rather than the specific methods used for training and optimization. The criteria used in our paper aim to evaluate how well features learned by SSL models capture invariances in the data, which remains valid regardless of the training divergence strategies. We appreciate your feedback and will emphasize this point in the revised manuscript.

---

> ### Comment · Reviewer_oq2z · 2024-11-19
> **Response for Authors rebuttal**
>
> Thank you for your reply. I don't think [1] and [2] only focus on training techniques for SSL tasks. Both of these works reveal key factors that can assist downstream tasks in pre-training contrastive processes. These two studies suggest that a good representation space should cluster data distributions by category. Alignment and enhancement implicitly introduce weak supervised signals. So, what I mean is that you should try to incorporate a measure of divergence for different latent class centers in your evaluation criteria. I believe this is the right direction.
>
> If there are any errors in my understanding, please feel free to correct me.

---

> > ### Author Response · Authors · 2024-11-19
> >
> > We sincerely thank the reviewer for their prompt response and constructive feedback. We greatly appreciate the opportunity to engage in this meaningful discussion and for the thoughtful suggestions that help refine our work.
> >
> > The central idea of [1] is to explore the trade-off between alignment and uniformity in learned representations, highlighting that simply increasing negative samples is not always advantageous. The authors propose strategies, such as improved sampling techniques, to optimize this balance, offering valuable guidance on the inclusion of negative samples during the training process. Similarly, [2] identifies alignment, divergence, and concentration as critical factors for the generalization ability of SSL tasks.
> >
> > Our work, however, focuses on interpreting well-trained SSL models to determine which regions are attended to during feature extraction, with the aim of providing human-understandable explanations. In contrast to [1] and [2], our primary objective is not to optimize SSL training or to analyze the sources of SSL task generalization ability. Nevertheless, these studies provide important insights and are highly relevant for guiding efforts to improve SSL generalization during training.
> >
> > We greatly appreciate your suggestion regarding divergence. Indeed, we have incorporated a similar consideration, albeit with a distinction: divergence evaluations typically require knowledge of class labels and downstream tasks. **As our work is specifically designed to remain independent of downstream tasks and labels,** this aspect falls outside the scope of our evaluation framework. As noted in [2], downstream task accuracy has a direct relationship with the divergence between latent class centers. To address this, we evaluated the impact of insertion and deletion on downstream tasks (refer to the table in our response to Weakness 4 from Reviewer 6yuv: [https://openreview.net/forum?id=2bEjhK2vYp&noteId=4sKAhovlWs]), where our method demonstrated significant improvements over IG.
> >
> > That said, we agree that directly evaluating the influence of individual features on divergence between latent class centers would be a meaningful addition. Given that our method is the first to directly interpret SSL tasks while introducing a novel and effective evaluation framework, we believe that extending this work to include such evaluations would be best pursued in future research.

---

> ### Comment · Reviewer_oq2z · 2024-11-25
> **Response for Authors rebuttal**
>
> Thanks to the authors for their patient explanations and excellent rebuttals.
>
> As you said, SSLA is specifically designed to remain independent of downstream tasks and labels. The lack of annotation information results in the non-computability of divergence. However, the purpose of recommending [1] and [2] is that both reveal NCE, Barlow Twins, and existing SSL methods have the capacity to optimize the divergence term, even though label information is not accessible. An alternative way to solve this issue is to figure out an upper bound for the divergence, the computation of which does not require any information about labels.
> Apart from that, the gap between the upstream and downstream tasks can be temporarily ignored, as the downstream dataset can be regarded as a subset of the pretraining dataset with some disturbed distribution shift when the pretraining dataset is comprehensive enough. Therefore, we can approximately think that the excellent structure of data distribution can be transferred to the downstream dataset. Thus, calculating the alternative quantity of divergence using pretraining data is acceptable.
> In summary, I am fairly certain that the motivation behind SSLA is valuable and deserves widespread attention. However, I don't think the authors sufficiently consider whether their core criterion is reasonable enough. I maintain that divergence is the most important factor in the success of current SSL methods, but the authors seem to ignore it.
>
> As stated above, I tend to maintain my score. If I have any misunderstandings, I would appreciate the authors pointing out my mistakes or engaging in a more in-depth discussion.

---

> > ### Author Response · Authors · 2024-11-25
> >
> > Thank you for your feedback.
> >
> > I completely agree with the point that these two papers, as well as the underlying rationale, explain the source of SSL generalization ability. However, as mentioned in the review, our goal is to explain what features SSL focuses on during the extraction process (i.e., which regions in the input samples are important) and that this importance is tied to the SSL task itself, rather than individual samples. Therefore, these two perspectives are not contradictory. If our work were about exploring the generalization performance of SSL tasks rather than explaining which regions SSL models focus on in samples, then the content of these two papers would inevitably require comparison and discussion.
> >
> > As we mentioned, we will incorporate this discussion. However, these two papers did not address (explaining which regions SSL models focus on in the samples) this task, so there is no conflict. Our work remains the first to achieve this goal.
> > It is also worth noting that in the original SSL papers, concepts such as divergence were not introduced, yet they still achieved excellent performance. This is another aspect that should be considered and subjected to interpretability analysis.
> >
> > Regarding "calculating the alternative quantity of divergence," we acknowledge that this has potential for future improvement of the work. However, it is not a necessary aspect to consider in this work. We cannot evaluate the quality of this work solely based on whether it can be improved in the future.
> >
> > I kindly hope you will reconsider the score based on these clarifications. Thank you again for your valuable insights and time.

---

> ### Comment · Reviewer_oq2z · 2024-11-25
> **Response for Authors rebuttal**
>
> Thank you for your further explanations.
>
> - I know the authors' goal is to explain the features SSL focuses on during the extraction process. However, the evaluation criterion adopted by the authors—cosine similarity—is, as I mentioned, not reasonable according to [1] and [2].
>
> - I am confident that the authors' work is valuable and that the studies in [1] and [2] do not address the same question as SSLA. I am simply suggesting that these studies should incorporate divergence into their evaluation criteria. While the original SSL papers do not introduce the concept of divergence, this is not a major issue. The initial motivation for a new method may differ from the essential factor that makes it successful, and this should not be a reason for the authors' rebuttal.
>
> - I have already provided a possible scheme for tackling the problem I mentioned. However, I agree with the author: we cannot evaluate the quality of this work solely based on whether it can be improved in the future. With that in mind, I am willing to improve my score to 5.

---

> > ### Author Response · Authors · 2024-11-25
> >
> > We understand your concerns and sincerely appreciate your willingness to adjust your score. The reason we chose cosine similarity as the evaluation criterion lies in its foundational role in SSL task design. In SSL training, similarity calculations are typically performed using normalized dot product similarity, which corresponds to cosine similarity. Our design aligns with the logic of attribution based on the loss functions used during training. This approach is reasonable, as it focuses on observing how the encoding of SSL outputs changes relative to the original encoding as the input samples vary.
> >
> > If divergence were incorporated into the training of SSL tasks, our evaluation framework could seamlessly adapt to use a divergence-based version. This mirrors the relationship between the original SSL papers and the works referenced in [1] and [2], where different perspectives can be used to discuss the same concept. Similarly, SSL tasks often evaluate performance by introducing downstream tasks to assess linear separability. The linear separability criterion is inherently consistent with the dot product operation used in SSL.
> >
> > We hope this clarification further justifies our choice of cosine similarity and its alignment with the principles underlying SSL task design. We sincerely appreciate your thoughtful feedback and your willingness to engage in such a meaningful discussion. Your insights have been invaluable in helping us refine our work.
> >
> > Given these clarifications and the alignment of our approach with SSL design principles, we kindly hope you might consider further improving the score, as your support would greatly encourage the ongoing development of this research. Thank you again for your time and consideration.

---

> > ### Author Response · Authors · 2024-12-01
> >
> > Dear Reviewer oq2z,
> >
> > Thank you for your review and feedback. As the rebuttal deadline approaches, we want to confirm that we have addressed the concerns you highlighted. If there are any additional questions or points to clarify, we are happy to provide further explanations.
> >
> > We respectfully request your reconsideration of the score.
> >
> > Best regards,
> >
> > Submission1920 authors

---

> > > ### Comment · Reviewer_oq2z · 2024-12-02
> > > **Response for Authors rebuttal**
> > >
> > > Thank you for your rebuttal.
> > >
> > > I don’t think the author has addressed my concern about the criterion in a convincing way; therefore, I tend to score 3 or 5.
> > >
> > > Considering that the author notice an important issue of getting rid of the influence of downstream tasks when considering interpretability, I keep my score at 5.

---

> > > > ### Author Response · Authors · 2024-12-02
> > > >
> > > > Dear Reviewer oq2z,
> > > >
> > > > Thank you for your feedback and for engaging in this meaningful discussion. We would like to seek further clarification on what you would consider a convincing justification for our evaluation criterion.
> > > >
> > > > Our choice of cosine similarity aligns directly with the design of SSL tasks, where normalized dot product similarity (equivalent to cosine similarity) is a foundational element used in training. This consistency ensures that our interpretability framework is tightly coupled with the logic and mechanisms inherent to SSL training processes. Could you clarify if this alignment poses any specific issues in your view?
> > > >
> > > > Our goal is to interpret SSL models in a manner consistent with how they are trained. By using the same distance metric employed during SSL task design, we ensure that our method faithfully reflects the model's internal logic and behavior. If this alignment is problematic, we would deeply appreciate further insights into how you believe the evaluation could be improved or what alternative would be more appropriate.
> > > >
> > > > Thank you once again for your valuable time and feedback. We look forward to your further clarification and hope this discussion can help refine the work even further.
> > > >
> > > > Best regards,
> > > >
> > > > Submission1920 Authors

---

> > > > > ### Comment · Reviewer_oq2z · 2024-12-02
> > > > > **Response for Authors rebuttal**
> > > > >
> > > > > Thank you for your rebuttal. My concerns about the divergence term have been specifically proposed in the above rebuttal. I believe that using only the alignment term as the evaluation criterion is inappropriate.

---

> > > > > > ### Author Response · Authors · 2024-12-02
> > > > > >
> > > > > > Given that SSL tasks often utilize Linear layers and KNN methods during downstream task evaluations, both of which rely on linear distance, we believe using alignment as the evaluation criterion is a natural and appropriate choice to reflect this aspect. Incorporating divergence into the evaluation process, while potentially valuable, would require extensive sampling and may introduce a degree of randomness. We kindly wonder if such a stochastic evaluation criterion would indeed be more suitable than the method we have proposed. While exploring this direction could be an excellent avenue for future work, we are confident that our current approach is both comprehensive and well-justified.

---

> ### Comment · Reviewer_oq2z · 2024-12-02
> **Response for Authors rebuttal**
>
> Why does evaluating the divergence term require extensive sampling and potentially introduce a degree of randomness?
>
> As far as the authors confidence. I can only acknowledge that you have the right to hold your own views, even though I have already explained that divergence is a crucial factor in the success of contrastive learning.

---

> > ### Author Response · Authors · 2024-12-02
> >
> > Thank you for your feedback and for raising your concerns regarding divergence.
> >
> > Referring to the original paper on divergence that you mentioned ([2] Weiran Huang, Mingyang Yi, Xuyang Zhao, and Zihao Jiang. "Towards the generalization of contrastive self-supervised learning." arXiv preprint arXiv:2111.00743, 2021), calculating divergence inherently requires the computation of class centers. This process involves sampling multiple instances to establish these centers, making the evaluation dependent on extensive sampling. In contrast, our evaluation method is designed to assess the model's interpretability using only the current sample, ensuring simplicity and reducing computational overhead.
> >
> > We fully acknowledge that divergence is a crucial factor in the success of contrastive learning. However, for explaining a successful SSL task, incorporating divergence is not a necessity. Given that SSL tasks have already demonstrated their effectiveness, our focus shifts to interpreting the critical regions attended to by these models. In this context, assessing the impact of important features on linear distances is the most suitable and direct approach.
> >
> > We hope this explanation clarifies our perspective, and we remain open to further discussion. Thank you again for your time and valuable insights.

---

> > > ### Comment · Reviewer_oq2z · 2024-12-02
> > > **Response for Authors rebuttal**
> > >
> > > Thank you for your response.
> > >
> > > Calculating divergence does not actually require the computation of class centers, as the class center is accessible in the upstream task, while the goal of SSLA is to ensure that the evaluation process does not depend on the downstream task. Therefore, as discussed above, determining an upper or lower bound for the class center based on the upstream dataset(s), whose computation does not rely on label information, is a reasonable approach, and this is also a concern that the author did not convince me of.
> > >
> > > In fact, these points have already been discussed in previous rebuttals, so there is no need to revisit them. When focusing on model interpretation, it is crucial to first clearly understand what is important for the model. Based on this, I believe the author's overall approach is correct, but the standards chosen are problematic.

---

> > > > ### Author Response · Authors · 2024-12-02
> > > >
> > > > Thank you for your feedback and for clarifying your perspective regarding the computation of divergence.
> > > >
> > > > We would like to reiterate that class centers cannot be directly accessed in scenarios where explicit class labels are unavailable. Although features can be obtained through input samples passed through the SSL model, without class labels, it is impossible to confirm whether these features represent intra-class dispersion or inter-class separation. This ambiguity makes the direct computation of divergence challenging.
> > > >
> > > > Furthermore, as the process requires sampling additional instances to estimate the divergence, the evaluation inherently involves sampling, which introduces a degree of randomness. While sampling can provide a reasonable approximation, the results would still depend on the specific samples chosen, adding variability to the evaluation process.
> > > >
> > > > We hope our clarification has addressed your concerns, and we kindly request you to reconsider your score in light of these explanations. Thank you for your continued engagement and valuable feedback.

---

> > > > > ### Comment · Reviewer_oq2z · 2024-12-02
> > > > > **Response for Authors rebuttal**
> > > > >
> > > > > Thank you for your rebuttal.
> > > > >
> > > > > However, I kindly suggest that the authors carefully read my previous responses above and refrain from revisiting past discussions, as they are meaningless for both of us. The current contrastive learning methods all propose alternative approaches (most of which are upper bounds of divergence) to address the incomputability of divergence. They do not require extra sample size to calculate, as the samples used to compute this term are the same as the samples used to calculate alignment, which is cosine similarity for SSLA.

---

> > > > > > ### Author Response · Authors · 2024-12-02
> > > > > >
> > > > > > Thank you for your feedback and valuable insights.
> > > > > >
> > > > > > We would like to clarify that our intention was not to revisit the same points but to address specific aspects raised in your comments. Based on the references you provided, incorporating divergence indeed involves sampling additional instances to compute meaningful estimates. If there are methods that do not require additional samples to calculate divergence, we would greatly appreciate it if you could share such references. This would be immensely helpful for us in improving our work in the future.
> > > > > >
> > > > > > Thank you again for your constructive suggestions and time.

---

> > > > > > > ### Comment · Reviewer_oq2z · 2024-12-03
> > > > > > > **Response for Authors rebuttal**
> > > > > > >
> > > > > > > Thank you for your rebuttal.
> > > > > > >
> > > > > > > Please refer to [1], especially Theorem 4. A similar effect can be achieved by [2] and [3]. This formula clearly does not require additional random sampling; just adopting the same samples used to calculate the cosine similarity is sufficient. In fact, despite adopting negative samples, additional samples are not required. For more details, please refer to https://github.com/jhaochenz96/spectral_contrastive_learning. There are lots of similar alternative plans, for example, dimensional contrastive mentioned in [4].
> > > > > > >
> > > > > > > [1] Weiran Huang, Mingyang Yi, Xuyang Zhao, and Zihao Jiang. "Towards the generalization of contrastive self-supervised learning." arXiv preprint arXiv:2111.00743, 2021
> > > > > > >
> > > > > > > [2] HaoChen, Jeff Z., et al. "Provable guarantees for self-supervised deep learning with spectral contrastive loss." Advances in Neural Information Processing Systems 34 (2021): 5000-5011.
> > > > > > >
> > > > > > > [3] Jeff Z. HaoChen, Colin Wei, Ananya Kumar, and Tengyu Ma. 2024. Beyond separability: analyzing the linear transferability of contrastive representations to related subpopulations. In Proceedings of the 36th International Conference on Neural Information Processing Systems (NIPS '22).
> > > > > > >
> > > > > > > [4] Garrido, Quentin, et al. "On the duality between contrastive and non-contrastive self-supervised learning." arXiv preprint arXiv:2206.02574 (2022).

---

> > > > > > > > ### Author Response · Authors · 2024-12-03
> > > > > > > >
> > > > > > > > Thank you for your feedback.
> > > > > > > >
> > > > > > > > The upper bound $\mu_k$ in Theorem 4 requires calculating the center point of class $k$, which depends on class annotations from downstream tasks. This calculation process clearly necessitates sampling from class $k$. Similarly, the work in [2] is derived based on specifically designed loss functions and strong assumptions, which are not feasible to compute in real-world scenarios (and can only provide a theoretical bound, meaning it cannot be practically evaluated). The approach in [3] also requires class labels to analyze the transferability between downstream tasks. [4], on the other hand, analyzes the difference between contrastive and non-contrastive methods, focusing on guiding SSL task design, which is significantly different from the evaluation of SSL interpretability. Therefore, we would like to ask whether there are divergence evaluation methods that do not rely on additional samples.

---

> > > > > > > > > ### Comment · Reviewer_oq2z · 2024-12-03
> > > > > > > > > **Response for Authors rebuttal**
> > > > > > > > >
> > > > > > > > > Thank you for your rebuttal.
> > > > > > > > > You should calculate $\mathcal{L}$ instead of $\mu_k$.

---

> > > > > > > > > > ### Author Response · Authors · 2024-12-03
> > > > > > > > > >
> > > > > > > > > > Thank you for your feedback.
> > > > > > > > > >
> > > > > > > > > > $\mathcal{L} _\text{infoNCE}$ requires sampling, whereas $\mathcal{L} _\text{align}$, like our method, uses linear distance but lacks angular information [1]. For instance, vectors such as [1, 2, 3] and [2, 4, 6] would have identical distances when evaluated for downstream classification tasks (which aligns with the foundational design principles of SSL). However, $\mathcal{L} _\text{align}$ would show a significantly different linear distance. In other words, relying solely on $\mathcal{L} _\text{align}$is insufficient. Given that $\mathcal{L} _\text{infoNCE}$ requires sampling, this does not align with the previously discussed perspective.

---

> > > > > > > > > > > ### Comment · Reviewer_oq2z · 2024-12-03
> > > > > > > > > > > **Response for Authors rebuttal**
> > > > > > > > > > >
> > > > > > > > > > > Thank you for your rebuttal.
> > > > > > > > > > >
> > > > > > > > > > > I earnestly request the author to carefully consider whether additional sampling is necessary to calculate $\mathcal{L}^{Cross}_2$ in Theorem 4. You can definitely use the samples for calculating cos similarity to calculate $L $in Theorem 4, right? Even if negative samples are used, they can still come from samples used to calculate cos similarity. Are we talking about the same thing?

---

> > > > > > > > > > > > ### Comment · Reviewer_oq2z · 2024-12-03
> > > > > > > > > > > > **Response for Authors rebuttal**
> > > > > > > > > > > >
> > > > > > > > > > > > For the implementation related to negative samples, refer to https://github.com/jhaochenz96/spectral_contrastive_learning

---

> > > > > > > > > > > > > ### Author Response · Authors · 2024-12-03
> > > > > > > > > > > > >
> > > > > > > > > > > > > Thank you for your reference.
> > > > > > > > > > > > >
> > > > > > > > > > > > > Please do not overlook our concern. The referenced work does not mention a specific evaluation standard or calculation method for divergence, nor does it even include the term "divergence" in the paper. We have carefully reviewed the papers you suggested, and we still have not found an evaluation method that meets the criteria you proposed.

---

> ### Comment · Reviewer_oq2z · 2024-12-03
> **Response to Authors rebuttal**
>
> The upper bound of divergence is an reasonable quantity to evaluate the divergence. Its calculation does not require extra samples.
>
> I have tried my best to indicate my concerns and provided a possible way to evaluate the divergence.
>
> but I don't think my concept can coincide with the authors' opinion.
>
> I decide to keep my score and accept AC's descision.

---

> > ### Author Response · Authors · 2024-12-03
> >
> > Thank you for your thoughtful comments and for taking the time to engage in this meaningful discussion. We respect your decision and remain grateful for the opportunity to exchange ideas.

---

### Official Review · Reviewer_CZeC · 2024-11-02

**Soundness:** 2
**Presentation:** 3
**Contribution:** 2
**Rating:** 3
**Confidence:** 4

**Summary:**

The paper proposes SSLA, a feature attribution method for self-supervised learning (SSL) tasks. In particular, the method is designed without dependency of downstream tasks. The method starts by defining the usefulness of SSL model as its ability to preserve representation of data after transformation. Then it addresses the significance of features by attributing this usefulness to features iteratively. The paper then conducts feature masking experiment to demonstrate the effectiveness of the method.

**Strengths:**

1. The paper proposed a novel feature attribution method for SSL model that does not rely on downstream tasks.
2. The claim made in the paper is supported by both theoretical derivations and experiments.
3. The paper is well written and easy to follow. The discussion of prerequisites of an attribution method for SSL may spark interesting discussions.

**Weaknesses:**

1. The paper relies on the independence of downstream tasks, which make the comparison of this method and existing methods difficult. Hence, it is difficult to address the effectiveness of this method.
2. The derivation of theorems rely heavily on first order approximation. Though it is common, the paper does not provide analysis on error bounds, which downgrades the trustworthiness of the method.
3. The two main components of the method lack motivation. The first one is using cosine-similarity of features before/after transformation as a measure of usefulness of SSL model. The correlation (or even causality) of this and "SSL as learning representation" is not clear. The second one is the iterative method. The author may consider justify why we need an iterative method to attribute the importance.
4. Although the paper proposes the method to be independent of downstream tasks, its evaluations still rely on downstream tasks, which seems counter-intuitive(Line 179 -180). Moreover, since the evaluation is dependent of downstream tasks, the author may consider compare their method to other SSL attribution methods that rely on downstream tasks.

**Questions:**

See weaknesses.

---

> ### Author Response · Authors · 2024-11-19
>
> ### Response for Weakness 1
> To address the difficulty in directly comparing SSLA with existing attribution methods, we provide extensive theoretical justifications and mathematical proofs to establish the validity and effectiveness of our method. These proofs rigorously demonstrate how SSLA aligns with established attribution axioms and resolves key challenges in SSL interpretability.
>
> ---
>
> ### Response for Weakness 2
> First-order approximations are a standard approach in influential attribution methods such as Integrated Gradients (IG) [1] and AGI [2], both of which have been widely validated. When the approximation is sufficiently close, the derived theories hold. In our work, we ensure the validity of the first-order approximation by setting a sufficiently low learning rate, minimizing approximation errors and enhancing the reliability of our method.
>
> ---
>
> ### Response for Weakness 3
> Cosine similarity is a core element of SSL training, as employed in methods like SimCLR [3] and MoCo [4]. It directly represents the essence of SSL, which involves learning invariant representations. Using cosine similarity aligns with SSL’s intrinsic properties, making it a natural and effective choice for our evaluation. Regarding the iterative approach, it ensures that the method satisfies attribution axioms, such as those outlined in IG [1]. Without iteration, the method would only capture local properties, akin to saliency maps, and fail to meet these axioms.
>
> ---
>
> ### Response for Weakness 4
> Our evaluation process does not rely on downstream tasks. Lines 179–180 provide definitions of typical SSL workflows to aid readers unfamiliar with SSL concepts, improving accessibility. However, these lines do not imply that our method requires downstream tasks for evaluation. Our experiments are designed specifically to avoid downstream task dependency, as detailed in the manuscript.
>
>
> Reference:
>
> [1] Sundararajan, Mukund, Ankur Taly, and Qiqi Yan. "Axiomatic attribution for deep networks." International conference on machine learning. PMLR, 2017.
> [2] Pan, Deng, Xin Li, and Dongxiao Zhu. "Explaining deep neural network models with adversarial gradient integration." Thirtieth International Joint Conference on Artificial Intelligence (IJCAI). 2021.
> [3] Chen, Ting, et al. "Simclr: A simple framework for contrastive learning of visual representations." International Conference on Learning Representations. Vol. 2. 2020.
> [4] He, Kaiming, et al. "MoCov1: Momentum Contrast for Unsupervised Visual Representation Learning." (2020): 9729-9738.

---

> > ### Author Response · Authors · 2024-12-01
> >
> > Dear Reviewer CZeC,
> >
> > Thank you for your thoughtful review. As the rebuttal deadline draws near, we have carefully addressed all the concerns you raised in your comments. If there are any remaining issues or further questions, we would be happy to address them.
> >
> > We humbly ask for your reconsideration of the score.
> >
> > Best regards,
> >
> > Submission1920 authors

---

> ### Comment · Reviewer_CZeC · 2024-12-01
>
> I sincerely thank the authors' response and am sorry for my late response.
>
> My concern on weakness #3 and #4 are addressed. On the other hand, I keep my concern that the derivation is not well supported by error bound analysis(weakness #2) and the effectiveness of the method is not well addressed because of a lack of comparison of existing methods. (weakness #1). Hence I currently keep my original score.

---

> > ### Author Response · Authors · 2024-12-02
> >
> > Dear Reviewer CZeC,
> >
> > Thank you for your feedback and for taking the time to review our work. As stated in our manuscript and rebuttal, to the best of our knowledge, our method is the first to address explainability in SSL tasks without relying on downstream tasks. We have not found similar methods for direct comparison. If you are aware of such methods, we kindly request that you provide examples to facilitate a meaningful comparison.
> >
> > Regarding traditional attribution methods, as mentioned in our response to Reviewer 6yuv (Response for Weakness 4), we have already compared our method with Integrated Gradients (IG), a well-established attribution method. Our results demonstrate that our method significantly outperforms IG in terms of capturing feature importance in SSL tasks.
> >
> > Given these clarifications and the novelty of our approach, we respectfully request that you reconsider your score.
> >
> > Best regards,
> >
> > Submission1920 Authors

---

### Official Review · Reviewer_bs4Q · 2024-11-03

**Soundness:** 2
**Presentation:** 3
**Contribution:** 2
**Rating:** 5
**Confidence:** 3

**Summary:**

This paper addresses the interpretability of SSL models, focusing on the challenge that existing interpretability methods often rely on downstream tasks or specific model architectures. To overcome these issues, the authors propose three fundamental prerequisites for SSL interpretability:
1. The interpretation should not introduce information from downstream tasks.
2. The interpretation process should not introduce samples other than the current sample.
3. The interpretation process should not be restricted to specific model architectures.

Based on these prerequisites, they introduce the Self-Supervised Learning Attribution (SSLA) algorithm. SSLA redefines the interpretability objective by introducing a feature similarity measure.
They also propose a new evaluation framework tailored to SSL tasks, arguing that traditional interpretability evaluation methods are impractical due to the absence of explicit labels and suitable baselines in SSL settings. Experiments are conducted using five representative SSL methods (BYOL, SimCLR, SimSiam, MoCo-v3, MAE) on the ImageNet dataset with ResNet-50 as the backbone. They compare SSLA against a random masking baseline, demonstrating that SSLA can more effectively identify important features that influence the SSL model's representations.

**Strengths:**

* **Novel Focus on SSL Interpretability:** The paper addresses an important and under-explored area—the interpretability of SSL models without reliance on downstream tasks or specific architectures.
* **Clear Prerequisites:** The authors clearly outline three prerequisites for SSL interpretability methods, providing a solid foundation for their approach.
* **Architecture-Agnostic Approach:** SSLA is designed to be independent of specific neural network architectures, potentially making it broadly applicable across different SSL models.
Recognition of Evaluation Challenges: The authors recognize the limitations of traditional interpretability evaluation methods in the context of SSL and attempt to propose a new framework tailored to SSL tasks.

**Weaknesses:**

* **Insufficient Empirical Evaluation:** The experimental evaluation is limited. The authors only compare SSLA to a random masking baseline. There are no comparisons with other existing attribution methods adapted to SSL, making it hard to gauge the effectiveness of SSLA.
* **Limited Dataset and Model Diversity:** Although experiments are conducted on the ImageNet dataset using ResNet-50, the evaluation lacks diversity in both datasets and model architectures. The claim that SSLA is architecture-agnostic is not fully supported without experiments on different architectures.
* **Evaluation Methodology Concerns:** The proposed evaluation framework is novel but not thoroughly validated. The authors argue that traditional evaluation methods are unsuitable for SSL interpretability but do not provide sufficient empirical evidence or theoretical justification. It is unclear whether the metrics used effectively measure interpretability in SSL contexts.

**Questions:**

*  **Comparison with Existing Methods:** Have you considered adapting existing attribution methods like Integrated Gradients or Grad-CAM to SSL settings for comparison? How does SSLA perform relative to these methods?
*  **Validation of Evaluation Framework:** How have you validated the effectiveness of your proposed evaluation framework? Have you conducted any studies or experiments to show that it correlates with human intuition or ground truth attributions?
*  **Testing on Diverse Architectures:** Given that experiments are only conducted with ResNet-50, have you tested SSLA on other architectures?
*  **Handling SSL Randomness:** How does SSLA account for the randomness inherent in SSL methods, such as stochastic data augmentations? Does this randomness affect the stability of the attribution results?
*  **Computational Overhead:** What is the computational cost of SSLA compared to standard inference? Is it feasible to apply SSLA to large-scale models and datasets?
*  **Generalization to Other Domains:** Can SSLA be applied to SSL models in domains other than computer vision?

---

> ### Author Response · Authors · 2024-11-19
>
> ### Response for Weakness 1
> As stated in Section 4.5, our method is the first to interpret SSL tasks without relying on downstream task information. This creates a lack of direct baselines for comparison. In this context, we designed experiments to validate the effectiveness of SSLA. Specifically, we evaluated five representative SSL methods (BYOL, SimCLR, SimSiam, MoCo-v3, and MAE), which cover major technical approaches in contrastive learning. This demonstrates the universality and effectiveness of SSLA. By comparing with a random masking baseline, we showcased SSLA's precision in distinguishing important and unimportant features. Furthermore, we innovatively employed cosine similarity as an evaluation metric, avoiding the baseline bias inherent in traditional insertion and deletion scores, as detailed in Section 4.3.
>
> ---
>
> ### Response for Weakness 2
> Using the ImageNet dataset and ResNet-50 aligns with experimental setups in existing interpretability methods, ensuring comparability and validation of our approach on widely recognized models. As shown in Algorithm 1, SSLA's architecture-agnostic nature stems from its algorithmic design, which abstracts away from specific model architectures. This abstraction makes SSLA applicable across different architectures, and its effectiveness is theoretically ensured through adherence to attribution axioms, as proven in Appendices B and C.
>
> ---
>
> ### Response for Weakness 3
> Traditional evaluation methods fail to address the interpretability of SSL tasks due to their reliance on baselines such as zero images or Gaussian blur, which are ineffective for SSL. SSL's focus on invariance, coupled with data augmentation already incorporating similar transformations, diminishes the impact of such baselines. Additionally, the subjectivity in baseline selection introduces bias, making these methods unsuitable for capturing the core mechanisms of SSL. As discussed in Section 4.3, we provided detailed theoretical justifications for our evaluation framework, supported by proofs in Appendix D.
>
> ---
>
> ### Response for Question 1
> Methods like Integrated Gradients and Grad-CAM require specified categories for their explanations, making them incompatible with SSL tasks. These methods are inherently tied to specific downstream tasks and cannot analyze SSL independently. Consequently, they are unsuitable for exploring the generalizable regions that SSL focuses on.
>
> ---
>
> ### Response for Question 2
> As noted in our response to Weakness 3, Section 4.3 and Appendix D contain detailed mathematical derivations and theoretical proofs validating the effectiveness of our proposed evaluation framework. To enhance reproducibility, we have uploaded visual examples to the rebuttal folder in the repository, which can be accessed via [https://anonymous.4open.science/r/SSLA-EF85/rebuttal/].
>
> ---
>
> ### Response for Question 3
> In addition to ResNet-50, we evaluated SSLA on models with architectures beyond ResNet-50. For example, MAE utilizes a ViT-based model, and our results in Table 1 show SSLA's strong performance when interpreting MAE. This highlights the generalizability of SSLA across architectures. Furthermore, our experiments covered major SSL methods, which are predominantly trained on ResNet-50 due to its effectiveness in SSL, as noted in foundational SSL papers.
>
> ---
>
> ### Response for Question 4
> SSLA inherently accounts for the randomness in SSL methods, such as stochastic data augmentations. As shown in lines 261–263 of the manuscript, we compute multiple evaluations and use the mathematical expectation to mitigate the impact of randomness, ensuring stable and reliable attribution results.
>
> ---
>
> ### Response for Question 5
> As summarized in the table below, SSLA requires only 51 forward and backward propagations, making it as efficient as IG [1] and significantly less computationally intensive compared to AGI [2] and ISA [3]. This efficiency makes SSLA feasible for large-scale models and datasets.
>
> | Method | Forward Passes | Backward Passes |
> |----------------|----------------|-----------------|
> | IG             | 51              | 51              |
> | AGI            | 1580            | 3144              |
> | ISA            | 512             | 512              |
> | **SSLA**       | **51**          |51              |
>
> ---
>
> ### Response for Question 6
> In principle, SSLA can be applied to SSL models in domains beyond computer vision. While our experiments focus on computer vision tasks, the theoretical foundation of SSLA is not domain-specific, enabling its extension to other domains with appropriate SSL setups.

---

> > ### Author Response · Authors · 2024-11-19
> >
> > Reference:
> >
> > [1] Sundararajan, Mukund, Ankur Taly, and Qiqi Yan. "Axiomatic attribution for deep networks." International conference on machine learning. PMLR, 2017.
> >
> > [2] Pan, Deng, Xin Li, and Dongxiao Zhu. "Explaining deep neural network models with adversarial gradient integration." Thirtieth International Joint Conference on Artificial Intelligence (IJCAI). 2021.
> >
> > [3] Zhu, Zhiyu, et al. "Iterative Search Attribution for Deep Neural Networks." Forty-first International Conference on Machine Learning.

---

> > ### Comment · Reviewer_bs4Q · 2024-11-29
> >
> > I thank the authors for their detailed and thoughtful response. While they have clarified some of my concerns, several fundamental issues remain unaddressed.
> > The main concern is the generalizability of their method and the lack of baseline comparisons. While it is possible to define a set of constraints for which no baselines exist, the key question is what practical value this brings. I believe the authors need to make a stronger effort to demonstrate the advantages and practical usefulness of their approach.
> > Based on these ongoing concerns, I maintain my original score.

---

> > > ### Author Response · Authors · 2024-11-30
> > >
> > > Thank you for the valuable response. As our work represents the first study in this specific area, it has indeed been challenging to identify comparable baselines.
> > >
> > > At the same time, exploring the explainability of SSL methods is of significant importance, particularly under the premise of not relying on downstream task information. SSL possesses high scalability and generalizability, making it a desirable approach for practitioners aiming to apply it across various downstream tasks.
> > >
> > > In this context, conducting explainability analyses of SSL methods without incorporating downstream tasks allows researchers to evaluate in advance whether the core regions of interest for downstream tasks are effectively captured by the model. If these regions are not adequately addressed, it indicates that SSL may not have fully leveraged the information from these regions during feature extraction. This approach can save substantial computational costs by avoiding repeated trial-and-error attempts. Therefore, analyzing SSL from the perspective of explainability not only helps researchers optimize the transferability of the model but also provides valuable guidance for the design and application of downstream tasks.
> > >
> > > We believe this line of research is highly insightful and practical, and we kindly ask you to reconsider the score.

---

> > > ### Author Response · Authors · 2024-12-01
> > >
> > > Dear Reviewer bs4Q,
> > >
> > > Thank you for your insightful comments. With the rebuttal phase nearing its conclusion, we have resolved the concerns you raised and provided detailed clarifications in our responses. Please let us know if there are any further questions.
> > >
> > > We kindly request you to reconsider your score.
> > >
> > > Best regards,
> > >
> > > Submission1920 authors

---

### Official Review · Reviewer_6yuv · 2024-11-04

**Soundness:** 2
**Presentation:** 3
**Contribution:** 3
**Rating:** 5
**Confidence:** 2

**Summary:**

The paper proposed a novel attribution algorithm for feature-level attribution on self-supervised learning. Compared to other feature-level attribution. The method is designed to meet prerequisites that the interpretation should not rely on 1) downstream task 2) other samples (other than the argumentation) and 3) model architectures. Authors present some experiments to justify the new method (SSLA) is effective.

**Strengths:**

- The prerequisites are designed to resolve the problem caused by other factors. And the method are design to reflect this spirit.
- The diagram are clear and help the readers to understand the method.

**Weaknesses:**

- I am not sure if the perquisites can be widely accepted by the community. For example, what is the downside (empirically / theoretically) if a downstream task is considered during the attribution process?
- Lack of comparison between different attribution methods. One interesting problem could be what's the difference between SSLA result v.s. other methods that relies on downstream tasks.
- Minor presentation suggestion
  - Equation 1 and 2 seems to be a little redundant.
- I am willing to raise the rate if the effectiveness of SSLA on some traditional evaluation methods (on downstream tasks) are proved (at least) to be correlated

**Questions:**

- In Figure 2, why we have a full R^n shape for a0, a1, ..., a_T?
- There seems to be no reason seperate the snowflake and light blue arrow in Figure 2?

---

> ### Author Response · Authors · 2024-11-19
>
> ### Response for Weakness 1
> The challenge of distinguishing whether attribution results derive from SSL itself or the downstream task highlights a critical issue. SSL demonstrates its strength by achieving superior performance across various downstream tasks, emphasizing its generalizability. Our approach focuses on understanding the origins of this generalizability. Specifically, by investigating which regions SSL prioritizes within samples, we aim to uncover why SSL methods excel across diverse downstream tasks rather than tailoring explanations to any single downstream task.
>
> ---
>
> ### Response for Weakness 2
> As mentioned at the beginning of Section 4.5, our method is the first to interpret SSL tasks without downstream task dependency, meaning there are no direct baselines for comparison. Traditional SSL interpretability methods inherently rely on downstream tasks, which makes them unsuitable for our context. Unlike these methods, our approach focuses on explaining which regions of the data SSL attends to, as these regions are the source of SSL's **generalizability**. Methods incorporating downstream tasks cannot explain this **generalizability**, making them inadequate for our study's objective.
>
>
> ---
>
> ### Response for Weakness 3
> Equations 1 and 2 are essential for establishing the attribution nature of our method. They serve as a foundation for proving that SSLA adheres to the Sensitivity Axiom and demonstrate how attribution results accumulate during sample updates. We will consider improving their presentation in future revisions to enhance clarity.
>
> ---
>
> ### Response for Weakness 4
> To address this concern, we conducted an evaluation using a linear classifier added to SSL, akin to typical SSL applications in downstream classification tasks. The Following Table results on the ImageNet dataset, comparing SSLA with the widely used Integrated Gradients (IG) [1] method. SSLA demonstrates significantly better performance on metrics such as Insertion (INS) and Deletion (DEL), indicating strong correlation and effectiveness in this setting.
>
> |      | INS    | DEL    |
> |------|--------|--------|
> | IG   | 0.0656 | 0.0125 |
> | SSLA | 0.2577 | 0.03   |
>
> Reference:
>
> [1] Sundararajan, Mukund, Ankur Taly, and Qiqi Yan. "Axiomatic attribution for deep networks." International conference on machine learning. PMLR, 2017.
>
> ---
>
> ### Response for Question 1
> The attribution results maintain the same dimensionality as the input because the process of explaining SSL tasks requires evaluating the importance of every input feature. This consistency ensures a one-to-one correspondence between the input and the attribution results.
>
> ---
>
> ### Response for Question 2
> The light blue arrows represent backpropagation, while the snowflake icon indicates computations that do not require gradient backpropagation. This distinction allows us to emphasize that these computations can be preprocessed independently, optimizing the overall computational efficiency of the attribution process.

---

> > ### Author Response · Authors · 2024-12-01
> >
> > Dear Reviewer 6yuv,
> >
> > Thank you for your valuable feedback on our submission. With the rebuttal deadline approaching, we would like to confirm that we have thoroughly addressed all the concerns you raised in your review. If there are any additional questions or issues, please let us know.
> >
> > We kindly request you to reconsider your score based on our clarifications.
> >
> > Best regards,
> >
> > Submission1920 authors

---

> > > ### Comment · Reviewer_6yuv · 2024-12-02
> > >
> > > Thank authors for the reply. I acknowledge I have read all the reviews and rebuttal replies. I have no more concerns about the paper. The only concern is the limited empirical improvement and weak baseline, this concern further developed to the possible practicality of this method. I will keep my rating as 5 for now.

---

> > > > ### Author Response · Authors · 2024-12-02
> > > >
> > > > Thank you for your follow-up comment. As we previously explained in our response to Reviewer bs4Q, our work represents the first study in this specific domain, which inherently poses challenges in identifying comparable baselines. Nevertheless, we firmly believe that exploring the interpretability of SSL methods, especially without relying on downstream task information, is crucial.
> > > >
> > > > SSL's scalability and generality make it an ideal framework for practitioners aiming to apply it across diverse downstream tasks. In this context, performing interpretability analysis on SSL methods without incorporating downstream tasks enables researchers to pre-evaluate whether a model effectively captures the core regions of interest for such tasks. If these regions are not sufficiently addressed, it may indicate that the SSL model has not fully leveraged the information from these regions during feature extraction. This approach can save significant computational costs by avoiding repetitive trial-and-error attempts.
> > > >
> > > > At the same time, as mentioned in our **Response for Weakness 4**, compared to traditional attribution methods such as IG, our approach demonstrates superior interpretability in SSL tasks. This is evident from its improved performance on metrics like Insertion (INS) and Deletion (DEL), which further validates the effectiveness and practicality of our method.
> > > >
> > > > From an interpretability perspective, analyzing SSL not only helps researchers optimize the model's transferability but also provides valuable guidance for the design and application of downstream tasks.
> > > >
> > > > Considering these clarifications, we kindly request you to reconsider your rating.

---

### Meta-Review · Area_Chair_vhYM · 2024-12-20

**Metareview:**

This paper introduces a novel interpretability framework for SSL models, aiming to decouple attribution from downstream tasks. The method, SSLA, uses a feature similarity measure to explain SSL models independently of downstream tasks or additional sample information. The paper outlines theoretical derivations, empirical results, and a new evaluation framework.

While the idea is innovative, several key concerns remain unresolved. Reviewers criticized the lack of baseline comparisons with existing attribution methods, even if those rely on downstream tasks. The reliance on cosine similarity as the sole evaluation metric was seen as insufficient, with Reviewer oq2z emphasizing the importance of divergence-based measures, which were not incorporated. Reviewer CZeC pointed out the reliance on first-order approximations without error bounds, weakening the theoretical robustness. Empirical validation was also limited, with experiments primarily conducted on ResNet-50 and ImageNet, raising questions about generalizability.

Although the authors actively engaged in the rebuttal phase, they did not convincingly address these concerns. I also agree with parts of the reviewers' concerns and think that the proposed method does not yet meet the bar for acceptance. I thus recommend rejecting the paper, but I encourage the authors to incorporate these helpful discussions into the revision and resubmit to a future venue.

**Additional Comments On Reviewer Discussion:**

The reviewers raised some concerns that remained unresolved through discussion. One issue was the reliance on cosine similarity as the sole evaluation metric. Reviewer oq2z argued that this was inadequate for assessing SSL models and emphasized the importance of incorporating divergence-based measures. While the authors defended cosine similarity as consistent with SSL design principles, they did not convincingly address alternative approaches suggested by oq2z, who maintained that divergence is central to understanding contrastive learning.

Another point was the lack of baseline comparisons. Reviewers bs4Q and 6yuv criticized the absence of tests against existing attribution methods, even if those rely on downstream tasks. The authors argued that SSLA’s independence from downstream tasks made direct comparisons inappropriate, but reviewers felt this weakened the empirical case for SSLA’s effectiveness.

Reviewer CZeC highlighted some theoretical issues, particularly the lack of error bounds in first-order approximations, which weakened confidence in the method’s robustness. The rebuttal did not fully resolve this concern.

While the authors actively engaged in the rebuttal phase, their responses failed to sufficiently address these core critiques. The reviewers thus were consistent in their view that the paper lacked the empirical and theoretical support needed for acceptance.

---

### Decision · Program_Chairs · 2025-01-22

Reject